# Nutraceuticals for the Treatment of Diabetic Retinopathy

**DOI:** 10.3390/nu11040771

**Published:** 2019-04-02

**Authors:** Maria Grazia Rossino, Giovanni Casini

**Affiliations:** 1Department of Biology, University of Pisa, via San Zeno 31, 56127 Pisa, Italy; rossinomariagrazia1@gmail.com; 2Interdepartmental Research Center Nutrafood “Nutraceuticals and Food for Health”, University of Pisa, 56124 Pisa, Italy

**Keywords:** retina, oxidative stress, inflammation, microvascular lesions, neoangiogenesis, polyphenols, flavonoids, carotenoids, saponins

## Abstract

Diabetic retinopathy (DR) is one of the most common complications of diabetes mellitus and is characterized by degeneration of retinal neurons and neoangiogenesis, causing a severe threat to vision. Nowadays, the principal treatment options for DR are laser photocoagulation, vitreoretinal surgery, or intravitreal injection of drugs targeting vascular endothelial growth factor. However, these treatments only act at advanced stages of DR, have short term efficacy, and cause side effects. Treatment with nutraceuticals (foods providing medical or health benefits) at early stages of DR may represent a reasonable alternative to act upstream of the disease, preventing its progression. In particular, in vitro and in vivo studies have revealed that a variety of nutraceuticals have significant antioxidant and anti-inflammatory properties that may inhibit the early diabetes-driven molecular mechanisms that induce DR, reducing both the neural and vascular damage typical of DR. Although most studies are limited to animal models and there is the problem of low bioavailability for many nutraceuticals, the use of these compounds may represent a natural alternative method to standard DR treatments.

## 1. Introduction 

Diabetic retinopathy (DR) is a retinal disease representing one of the main causes of vision loss in developed countries. It has been classically considered a microvascular disease of the retina and is characterized, in its later stages, by abnormal growth of retinal vessels, which causes hemorrhages and tractional retinal detachment, leading to vision loss [1]. The understanding of DR has evolved over time and has clarified the role of the neuronal component of the retina in the progression of the disease. Indeed, growing experimental evidence suggests that suffering and death of retinal neurons occur before overt vascular changes [2,3,4]. For this reason, nowadays DR can be described not only as a microvascular but also as a neurodegenerative disease of the retina [5].

DR is a multifactorial disease but, to date, the exact pathophysiological mechanisms underlying neuro-vascular damage are not thoroughly understood. Nevertheless, different pathways and molecular mechanisms that may cause DR onset have been studied. For instance, the increase in advanced glycation end-products (AGEs) acting at their receptors (RAGE), the formation and activation of protein kinase C (PKC), or the increased flux in the polyol or hexosamine pathway have been examined [6,7]. All of these pathways, along with lower levels of glutathione (GSH), are associated with an increase in oxidative stress. The latter in turn causes different alterations in the diabetic retina as a consequence of severe lipid peroxidation, protein oxidation, oxidative DNA damage, induction of inflammation, and upregulation of growth factors, such as vascular endothelial growth factor (VEGF) [8].

VEGF is a proangiogenic factor that plays a key role in the late vasculopathy. For this reason, current DR treatments consist of the intraocular delivery of anti-VEGF molecules whose action induces restriction or inhibition of abnormal vessel growth. Nevertheless, the administration of anti-VEGF drugs has limitations and may generate different side effects. In addition, the effects are not long-lasting and frequent intravitreal injections are necessary [9,10,11].

Recent studies have highlighted the neuroprotective role of VEGF that can be noticed in the early phases of DR [12,13]. According to these studies, retinal neurons stressed by diabetes are likely to trigger the release of VEGF as a survival strategy. However, the persistence of the upstream stress conditions determines the accumulation of VEGF, leading to disruption of the blood-retina barrier (BRB) and, in the long term, to neoangiogenesis [14]. It would therefore be appropriate to plan new therapeutic strategies acting upstream of the disease and to prevent its progression by reducing neuronal stress and favoring neuroprotection. Moreover, considering the side effects caused by therapeutic agents administered via intraocular injections, there is a need to develop compounds with antioxidant and/or anti-inflammatory activity that can be administered through alternative delivery modalities. For this reason, in the last few years, several studies have focused on the potential benefits of nutraceuticals.

The term “nutraceutical” was coined by Dr. Stephen De Felice in 1989 and indicates “a food (or part of a food) that provides medical or health benefits, including the prevention and/or treatment of a disease” [15]. Nutraceuticals are effective antioxidants. They may induce the expression of antioxidant enzymes, act as scavengers of reactive oxygen species (ROS), or display singlet oxygen-quenching activity, as in the case of carotenoids [16]. Nutraceuticals may also exert anti-inflammatory effects by reducing the expression or nuclear translocation of nuclear factor kappa-light-chain-enhancer of activated B cells (NF-κB) [17]. Nutraceuticals can be used as natural dietary supplements and therefore can be easily administered, are readily available, and are affordable. A further advantage of nutraceuticals is that they are not likely to induce collateral side effects (if, of course, delivered at the appropriate dosage) such as hypoglycemia, liver injury, or gastric complains, which are characteristic of well-known and popular drugs [18,19].

In this review, we focus our attention on different classes of nutraceuticals, such as polyphenols, carotenoids, saponins, and others (Figure 1), explaining how these substances might counteract DR pathological changes. In particular, we highlight how nutraceuticals may reduce (i) oxidative stress; (ii) inflammation; (iii) neurodegeneration; and (iv) vascular changes. Reviewed literature includes in vitro studies, in vivo studies on animal models, and also clinical studies. Finally, we also consider how the low bioavailability of several nutraceuticals may limit their use.

## 2. Nutraceuticals and Oxidative Stress

Oxidative stress is caused by an imbalance in the production of ROS and the activity of the biological detoxifying systems. ROS are produced in normal metabolic conditions to support normal cellular functions and modulate a variety of biological processes including cell proliferation, differentiation, and migration, signal transduction, and programmed cell death [8]. However, because of ROS’ high reactivity, their accumulation compromises the cell structure and functionality through alterations and degradation of molecules such as DNA, lipids, and proteins [20]. Oxidative stress and ROS production are contrasted by endogenous antioxidant defense enzymes including superoxide dismutase (SOD), catalase (CAT), glutathione peroxidase (GSH-P), and glutathione reductase (GSH-R). In addition to these endogenous enzymatic systems, endogenous non-enzymatic factors also exist and they include GSH (which is regulated by GSH-P and GSH-R), vitamin C, and vitamin E [21]. Besides endogenous antioxidant defenses, exogenous antioxidants of natural origin may be used to preserve redox homeostasis. They may act directly as scavengers of free radicals, indirectly by interrupting free radical chain reactions, or both. They may also decrease oxidative stress by inducing the expression of endogenous antioxidant enzymes [22,23]. For these reasons, it has been recently proposed that therapies based on natural, non-enzymatic antioxidants such as nutraceuticals could relieve the decrease in endogenous antioxidant defenses [23,24].

The retina is highly susceptible to oxidative stress, which is due principally to the high content of polyunsaturated fatty acids, high oxygen uptake, glucose oxidation, and prolonged exposure to light. In particular, high glucose levels trigger a set of processes, such as AGE accumulation, PKC activation, and increased flux in the polyol and hexosamine pathways, which provoke oxidative stress (see [25] for detail). In turn, an increase in ROS is likely to cause DNA fragmentation resulting in poly-ADP ribose polymerase activation and glyceraldehyde 3-phosphate dehydrogenase inhibition [26]. This causes accumulation of glycolytic metabolites that may induce AGE formation and activation of PKC and of the polyol as well as of the hexosamine pathways, which are known to contribute to DR pathogenesis [6,7]. In summary, oxidative stress creates a propagating cycle, causing a continuous increase in ROS and consequent activation of pathways closely related to the progression of DR [8] (Figure 2).

Different natural dietary compounds have been investigated as possible treatments or adjuvants to counteract retinal oxidative stress typical of DR. They include polyphenols, carotenoids, and saponins, as well as other compounds (Figure 1). They are common in different fruits, vegetables, herbs, and beverages, and are very efficient in strengthening the endogenous antioxidant defenses through a direct scavenger activity and/or through the stimulation of antioxidant enzyme expression. Several classes of these compounds have been tested in vitro and in in vivo animal models. A summary of the effects of nutraceuticals against oxidative stress in models of DR is given in Figure 3.

### 2.1. Non-Flavonoid Polyphenols

Curcumin, a yellowish polyphenolic substance constituting the major active compound of *Curcuma longa*, is widely known for its antioxidant and anti-inflammatory properties [27,28,29]. The strong antioxidant power of curcumin has been shown in different studies. In human retinal endothelial cells (HRECs) exposed to high glucose and treated with 10 μM curcumin, intracellular ROS production has been observed to be significantly reduced [30], and similar results have been obtained with the retinal pigment epithelial cell line ARPE-19 [31,32]. The decrease in ROS levels is concomitant with an increased expression of heme oxygenase-1 (HO-1) [31], a redox-sensitive inducible stress protein that, once activated, protects the cell from different types of stress. This observation suggests that curcumin not only generates direct antioxidant activity but it may also act indirectly by enhancing the expression of antioxidant enzymes. This effect is likely to be induced through activation of the transcription nuclear factor erythroid-2-related factor-2 (Nrf2). Once activated, Nrf2 translocates into the nucleus and promotes the transcription of genes that encode antioxidant enzymes (known as phase II antioxidant enzymes), including HO-1 [33,34]. Another recent investigation into high glucose-stressed ARPE-19 cells showed that curcumin-induced inhibition of ROS formation prevents alterations of DNA methyltransferase activity [35]. In in vivo studies with rats with streptozotocin (STZ)-induced diabetes, curcumin has been observed to prevent the retinal increase of malondialdehyde (a marker of oxidative stress) and the decrease in GSH [36]. In the same model, curcumin also inhibited a decrease in total antioxidant capacity by increasing SOD, CAT, and GSH levels [37,38] and prevented an increase in the levels of retinal nitrotyrosine, a marker of oxidative protein damage, and in 8-hydroxy-2′-deoxyguanosine, a marker of oxidative DNA damage [38].

Among other non-flavonoid polyphenols, resveratrol, found in different plants such as grapes, peanuts, and berries, has been described as being able to decrease oxidative stress in retinas of diabetic rats by reducing lipid peroxidation, oxidized to reduced GSH ratio, and superoxide dismutase activity [39]. Recent data also show that resveratrol may reduce the adverse effects of hyperglycemia-induced oxidative stress on retinoic acid metabolism, which is involved in the recycling of 11-cis-retinal in the visual cycle in the retinal pigment epithelium [40].

### 2.2. Flavonoid Polyphenols

Flavonoids, a class of polyphenols, constitute a variegated group of natural substances characterized by strong antioxidant power. These natural products are present in fruits, vegetables, grains, roots, tea, and wine [41]. In STZ diabetic rats, treatment with different flavonoids ameliorates retinal redox status favoring an increase in GSH and a decrease in lipid peroxidation. It has also been observed that flavonoids are able to increase the levels of antioxidant enzymes such as SOD and CAT. In particular, these findings have been recorded in retinas of diabetic rats treated with quercetin, a common flavonol found in vegetables and fruits [42], with hesperetin, a flavanone commonly present in citrus fruits [43], or with green tea [44]. Green tea is a popular beverage rich in catechin, epicatechin, epigallocatechin, epicatechin gallate, and epigallocatechin gallate. Among these, epigallocatechin gallate is the most abundant catechin in green tea and is widely known for its antioxidant activity. The antioxidant effect of epigallocatechin gallate seems to be associated with a decrease in aldose reductase activity, which catalyzes the rate limiting step in the polyol pathway, and a decrease in AGE accumulation [45].

Eriodictyol, a flavonoid extracted from yerba santa (*Eriodictyon californicum*), a plant native to North America, has been found to reduce ROS production and increase the activity of SOD, GSH-P, and CAT. In addition, it has been shown to enhance the nuclear translocation of Nrf2 and elevate the expression of antioxidant enzyme HO-1 in RGC5 cells treated with high glucose [46].

Anthocyanins constitute another class of flavonoids which are responsible for the red or blue color of plants, fruits, and flowers. In vitro studies with HRECs subjected to high glucose treatment have shown that the blueberry anthocyanins malvidin and malvidin glycoside may produce an antioxidant effect through reduction of ROS levels and an increase in both CAT and SOD activity [47]. In addition, blueberry anthocyanins added to the food of and administered to diabetic rats for 12 weeks have been described as being able to prevent retinal oxidative stress favoring an increase in antioxidant capacity, as demonstrated by an increase in GSH and decrease in ROS levels. This antioxidant activity of blueberry anthocyanins is mediated by activation of Nrf2 and a consequent increase in HO-1 expression [48]. 

### 2.3. Carotenoids

Lutein and zeaxantin are the principal constituents of oranges, yellow fruits, and dark green leafy vegetables. Together with *meso*-zeaxanthin, they form the macular pigment of primate eyes [49] and prevent oxidative damage to the retina [50,51]. Their potential role in protecting against visual disorders has been recently reviewed [52,53,54]. Regarding the effects of carotenoids in experimental models of DR, a decrease in lipid peroxidation, nitrotyrosine levels, and oxidatively modified DNA was observed in the retinas of diabetic rats that had received supplementation with zeaxantin for two months. These effects were accompanied by inhibition of the diabetes-induced decrease in retinal SOD expression and activity, although no effects of zeaxantin were observed on GSH levels [55]. Similarly, lutein administration to one-month-diabetic mice has been observed to prevent retinal oxidative stress and restore retinal ROS levels to normal [56].

Crocetin and crocin are two additional compounds belonging to the class of carotenoids. They can be considered as the active ingredients of saffron, a spice classically used in traditional medicine for its beneficial qualities [57]. Crocetin and crocin, similar to lutein and zeaxantin, are known for their antioxidant and protective actions against ROS. For instance, crocetin has been reported to protect cells of the RGC5 cell line from oxidative stress [58], while treatment with crocin has been shown to prevent upregulation of ROS and nitric oxide in microglia cells cultured in high glucose [59].

### 2.4. Saponins

*Panax notoginseng* saponins (PNS), including ginsenoside Rg1, ginsenoside Rb1, and notoginsenoside R1, may generate a protective effect against oxidative stress-induced damage, as observed in STZ diabetic mice treated for two months with North American Ginseng (*Panax quinquefolius*) [60]. In addition, a significant decrease in ROS levels has been recorded in rat retinal capillary endothelial cells exposed to high glucose and treated with 100 μg/mL of PNS [61]. This decrease in ROS levels has been associated with an increase in antioxidant enzymes, including SOD, CAT, and, consequently, GSH. In addition, notoginsenoside R1 was also observed to induce a decrease in the activity of NADPH oxidase, the major enzyme implicated in oxygen radical generation [62].

### 2.5. Other Compounds

Lisosan G is a fermented powder obtained from organic whole grains (*Triticum aestivum*). It is enriched in bioactive substances such as phenolic components, flavonoids, alpha-lipoic acid, tocopherols, and polyunsaturated fatty acids (see [63] for detail). In mouse retinal explants, Lisosan G has been shown to inhibit an oxidative stress-induced increase in phase II antioxidant enzymes such as HO-1, SOD, and glutamate-cysteine ligase catalytic subunit mRNA expression, while in STZ rats it was observed to seem to inhibit the nuclear translocation of Nrf2, indicating that in these systems Lisosan G is likely to exert antioxidant effects through direct radical scavenging and not through activation of antioxidant enzyme expression [63].

## 3. Nutraceuticals and Inflammation 

Inflammation is a nonspecific response to injury that includes a variety of functional mediators, such as cytokines, chemokines, acute phase proteins, and other pro-inflammatory molecules. Many of these mediators have been detected in the retina of diabetic animals or patients, suggesting that inflammation has a role in the development of DR [64,65,66]. Reactive gliosis, characterized by increased glial fibrillary acidic protein (GFAP) expression in both Müller cells and astrocytes [4,67], is typically observed in DR [68,69], resulting in the release from these cells of inflammatory cytokines, such as tumor necrosis factor alpha (TNFα), interleukin 1 beta (IL-1β), and others [70,71]. 

The transcription of inflammatory proteins is regulated by the activation of pro-inflammatory transcription factors, among which NF-κB plays a prominent role. This factor, once activated, translocates into the nucleus, binds to nuclear DNA, and acts as a master switch that promotes the expression of pro-inflammatory cytokines such as IL-1β, interleukin 6, interleukin 8 (IL-8), and, at least in part, TNFα [72]. There is ample evidence suggesting that NF-κB is involved in the pathogenesis of the early phases of DR. In fact, the inhibition of proteins whose expression is regulated by NF-κB decreases capillary degeneration, while direct NF-κB blockade inhibits DR development and progression [64,65,66,73]. The potential efficacy of some nutraceuticals for the treatment of DR is that they may inhibit NF-κB activation. A summary of the effects of nutraceuticals against inflammation in models of DR is given in Figure 3.

### 3.1. Non-Flavonoid Polyphenols

Treatment with curcumin prevents TNFα release in HRECs cultured with high glucose [32]. Curcumin also reduces retinal diabetic damage in diabetic rats through inactivation of NF-κB and a decrease in IL-1β levels [38]. There is some indication that curcumin may influence NF-κB by preventing the diabetes-induced retinal activation of calcium/calmodulin-dependent protein kinase II (CAMKII) [27,74,75,76], a ubiquitous multifunctional protein kinase implicated in the regulation of the transcriptional activity of NF-κB [77]. Curcumin has also been observed to reverse the diabetes-induced upregulation of retinal GFAP in Müller cells of STZ rats [36].

Recent observations have shown that inflammatory markers are reduced in the retinas of STZ rats after administrations of resveratrol via tail vein injections [78]. Similar to curcumin, resveratrol, or an ethanol extract of the root of *Polygonum cuspidatum*, which is rich in resveratrol, attenuates inflammation in the retinas of diabetic rats by reducing NF-κB activity [39,79]. In addition, resveratrol has been described as being able to reduce NF-κB nuclear translocation in the retinas of mice with experimental uveitis [80]. Resveratrol is likely to promote the inhibition of NF-κB through AMP-activated protein kinase (AMPK) activation. Indeed, data obtained from the retinas of mice with STZ-induced diabetes has shown that resveratrol-induced AMPK activation leads to significant suppression of NF-κB phosphorylation and reverses diabetes-induced sirtuin-1 (SIRT1) deactivation [81]. This SIRT1 activation promoted by resveratrol is likely to mediate an inhibition of NF-κB stimulation of DNA transcription, since SIRT1 deacetylates both NF-κB p65 and histone 3, with the effect of decreasing DNA binding by NF-κB [82]. Similarly to curcumin, a mechanism by which resveratrol may negatively modulate NF-κB is the inhibition of retinal CAMKII activation [83].

### 3.2. Flavonoid Polyphenols

Quercetin displays both antioxidant and anti-inflammatory properties in the retina. In particular, it has been reported to reduce VEGF-induced inflammation by inactivating NF-κB through inhibition of both mitogen-activated protein kinase and Akt in 661W cells [84]. In STZ diabetic rats, quercetin inhibits an increase in retinal GFAP expression and induces a decrease in NF-κB protein expression in specific retinal layers, namely the nerve fiber layer, the inner plexiform layer (IPL), and the inner nuclear layer (INL). This effect of quercetin on NF-κB is also associated with decreased levels of TNFα and IL-1β [42].

Hesperetin is another flavonoid that has been reported to exert antioxidant effects in diabetic retinas, as reported above. This compound has also been observed to inhibit the diabetes-induced over-expression of GFAP and of the pro-inflammatory cytokines TNFα and IL-1β in retinas of diabetic rats [43]. Eriodictyol, a flavonoid of the same class of hesperetin, has been reported to also reduce TNFα in STZ rat retinas [85] or both TNFα and IL-8 in high glucose-stressed RGC-5 cells [46].

Catechin has been observed to increase heat shock protein 27 levels and decrease the production of associated inflammatory factors in retinas of STZ rats [86]. Diabetes induced glial activation in the retina, characterized by increased GFAP expression in Müller cells, has also been found to be inhibited by green tea or by epicatechin [87,88].

### 3.3. Carotenoids

Among the carotenoids, crocin has been observed not only as being able to protect from oxidative stress, but also to block the pro-inflammatory response in microglial cells challenged with high glucose and free fatty acids. In both the antioxidant and the anti-inflammatory action of crocin, activation of the phosphoinositide 3-kinase (PI3K)/Akt signaling seems to play a significant role [59].

### 3.4. Other Compounds

The compounds 6-gingerol and the sesquiterpene zerumbone are abundantly present in rhizomes of the plants of the ginger family *Zingiber officinale* and *Zingiber zerumbet*, respectively. They are able to ameliorate retinal damage induced by hyperglycemia by inhibiting NF-κB expression/activation and reducing the expression of pro-inflammatory cytokines [89,90]. In particular, the effect of zerumbone is likely to be due to the blockading of the AGE/RAGE/NF-κB pathway [90].

Lisosan G has been reported to block increases in GFAP mRNA expression, indicating reactive gliosis, induced by diabetes in the retinas of STZ rats [63]. In addition, Lisosan G has been shown to exert important anti-inflammatory effects that can be associated with a reduction in NF-κB nuclear translocation, as observed in hepatocytes or in human endothelial progenitor cells [91,92] and has recently been hypothesized in an in vivo rat model of DR, where a Lisosan G-induced reduction of NF-κB phosphorylation was reported [63].

A post-translational modification (O-GlcNAcylation) of NF-κB has been observed in several pathologies, including DR [93]. An extract of *Aralia elata* (a plant traditionally used to treat diabetes in Eastern countries) containing phenolic compounds (3, 4-dihydroxybenzoic acid, chlorogenic acid, and caffeic acid) has been recently shown to reduce glial activation, to suppress NF-κB expression, and to decrease its O-GlcNAcylation in the retinas of STZ diabetic mice [94]. Finally, a fortified extract of red berries, *Ginkgo biloba*, and white willow bark containing carnosine and α-lipoic acid have been reported to attenuate the increase in TNFα levels in the retinas of STZ rats [95].

### 3.5. Relationships between Inflammation and Oxidative Stress

It is interesting to observe that most of the compounds cited above display both antioxidant and anti-inflammatory properties, as has been reported in different in vitro and in vivo experimental models. In fact, oxidative stress has been recognized as playing a pivotal role in the development of inflammation [96,97]. Accordingly, ROS production is likely to promote activation of NF-κB, an oxidant-sensitive factor and a crosslink between inflammation and oxidative stress [98,99,100]. Recent evidence in ARPE-19 cells also suggests that high glucose-induced ROS may promote the secretion of inflammatory cytokines through PI3K/Akt/mTOR, and curcumin has been found to inhibit this signaling pathway [101]. In summary, in DR inflammation is likely to be secondary to increased oxidative stress, and the use of appropriate antioxidant compounds may prevent the establishment of an inflammatory state.

## 4. Nutraceuticals and Neurodegeneration

DR is characterized by an extended loss of neurons due to an increase in apoptosis likely paralleled by a decrease in autophagic capabilities [102]. Neuronal cell vulnerability is evident very early in DR, and it is detectable before any sign of vascular damage [2,3,4]. This early neuronal impairment leads to retinal functional deficits that can be recorded with electroretinography (ERG) and that are associated with different morphological changes, these mostly including a decrease in thickness of retinal layers, with INL and IPL affected in particular. In retinas of diabetic rodents, an increase in terminal deoxynucleotidyl transferase-mediated dUTP nick end labelling (TUNEL) positive cells can be recorded together with a decrease in anti-apoptotic markers (e.g., B cell lymphoma 2 (Bcl-2)) and an increase in pro-apoptotic markers (e.g., active caspase-3 and Bcl-2-associated X protein (Bax)) [103,104,105]. Neurodegeneration in DR is likely caused by high glucose-induced oxidative stress and inflammation, but there is evidence that dysregulation of neurotrophic factor expression may also play a role. Neurotrophin nerve growth factor (NGF) and brain-derived neurotrophic factor (BDNF) are expressed by retinal neurons and glia, and are principally involved in cell survival and synaptic modulation [106,107]. A reduction in neurotrophin expression or an imbalance between the mature neurotrophin and its precursor (as in the case of proNGF/NGF) may lead to neuronal damage and neurodegeneration [107,108]. A further cause of neuronal death in DR is represented by increased glutamate levels causing excitotoxicity. This condition is likely to be due to oxidative stress in Müller cells resulting in decreased activity of glutamate-aspartate transporters and down-regulation of glutamine synthetase (GS), which converts glutamate into non-toxic glutamine [109].

Several natural compounds are known for their neuroprotective properties and for their positive effects within the central nervous system. In particular, nutraceuticals rich in flavonoids have been proposed for the treatment and prevention of a variety of neurodegenerative diseases [110,111]. A summary of the effects of nutraceuticals against neurodegeneration in models of DR is given in Figure 3.

### 4.1. Non-Flavonoid Polyphenols

In diabetic rat retinas, curcumin has been reported to exert antiapoptotic effects by upregulating the expression of Bcl-2 and downregulating the expression of Bax, with reduction of apoptosis of retinal ganglion cells and of cells in the INL and preservation of normal retinal thickness [112]. In addition, curcumin reverses diabetes-induced down-regulation of retinal GS, which may aid glutamate clearance and reduce the risk of excitotoxicity [36]. Interestingly, curcumin may also contribute to inhibition of apoptosis by promoting autophagic flux in retinal neurons. Indeed, curcumin has been reported to stimulate autophagy and exert protective effects in different models of central nervous system neurodegeneration [113].

Resveratrol has been shown to reduce retinal apoptotic levels and attenuate retinal thinning in rats with STZ-induced diabetes [39,78]. The neuroprotective action of resveratrol is likely to be associated with its anti-inflammatory action [83]. Similarly to curcumin, resveratrol may inhibit apoptosis by stimulating autophagy. Indeed, it has been reported to induce autophagy and reduce cell death both in the human retinal pigment epithelial ARPE-19 and in mouse photoreceptor 661W cells exposed to cytotoxic stress [114].

### 4.2. Flavonoid Polyphenols 

The beneficial effects of flavonoids on retinal neurodegeneration in DR have been the subject of numerous studies. Treatment of STZ rats with quercetin protects from diabetes-induced retinal ganglion cell loss, mitigates thinning of retinal layers, reduces caspase-3 expression/activation and the levels of cytochrome c, while increasing Bcl-2 [42,115]. In addition, quercetin improves the expression of neurotrophic factors, of their receptors, and of their downstream signaling molecules. In particular, quercetin treatment favors an increase in Akt phosphorylation and in the expression of BDNF, its receptor Trk-B, and in synaptophysin. These data suggest that the neuroprotective action of quercetin is mediated by the BDNF-Trk-B/Akt-synaptophysin pathway [115]. The possibility that quercetin may affect apoptosis through the promotion of autophagy is supported by observations reporting potent simulation of autophagy by quercetin in Schwann cells with high glucose [116] and quercetin protection from Aβ-induced neurotoxicity through the induction of autophagy in *C. elegans* [117]. Similarly to quercetin, the flavonol kaempferol, which is found in tea, broccoli, apples, strawberries, and beans [118], could also be a stimulator of autophagy, as demonstrated in the human neuroblastoma SH-SY5Y cell line [119].

Another flavonoid that could represent a good choice with which to counteract neurodegeneration in DR is rutin. It is the main glycoside form of quercetin and is abundant in foods such as onions, apples, tea, and red wine [120]. Its neuroprotective effects have been tested in rat retinal ganglion cells subjected to oxidative stress, where treatment with rutin was observed to increase cell survival rate and reduce caspase-3 activation [121]. Rutin anti-apoptotic action has been confirmed in diabetic rats, in which treatment with this compound was observed to cause a decrease in caspase-3 activity and expression, with a concomitant increase in Bcl-2 and preservation of the levels of both BDNF and NGF [122].

Chrysin, a natural flavonoid found in herbs and honeycomb, has been recently shown to protect retinal photoreceptors by maintaining robust retinoid visual cycle-related components in glucose-stimulated human retinal pigment epithelial cells or in the retinal pigment epithelium of diabetic rats [123].

The strong antioxidant power of hesperetin correlates with the neuroprotective actions of this flavonoid. In diabetic retinas, it inhibits neuronal death, reducing caspase-3 expression [43], and prevents retinal thinning, favoring protection of ganglion cells and of cells in the INL [124]. Naringenin, also found in citrus fruits together with hesperetin, exerts similar neuroprotective actions in diabetic retinas, favoring an increase in BDNF and synaptophysin together with reduction in apoptotic levels, as indicated by increases in Bcl-2 and decreases in both Bax and caspase-3 expression [125]. The naringenin-promoted decrease in pro-apoptotic molecules is likely to be due to activation of Akt and Erk 1/2, as shown in hippocampal cells subjected to excitotoxic stress and treated with different concentrations of naringenin [126]. Anti-apoptotic effects of the flavanone eriodictyol have also been reported in high glucose-stressed RGC-5 cells [46].

Other studies have shown that treatment with green tea may prevent neurodegeneration in diabetic retinas. Indeed, the oral administration of green tea to diabetic rats generates a neuroprotective action in the retina characterized by a reduction in neuronal death, restoration of glutamate uptake, and improvement of retinal functionality as recorded with ERG [87]. The neuroprotective effect of epicatechin in retinas of diabetic rats has been proposed to be related to the reduction of pro-NGF production [88].

### 4.3. Carotenoids

Lutein is the carotenoid with the most recognized neuroprotective effects in the diabetic retina. Its constant intake induces an evident functional improvement, as highlighted by an ERG analysis of oscillatory potentials in the retinas of diabetic mice, which indicates prevention of inner retinal damage [127]. Moreover, lutein treatment restores retinal layer thickness, reduces retinal apoptosis, and preserves both BDNF and synaptophysin levels [128]. Lutein and zeaxantin are present in *Lycium barbarum*, a shrub member of the family Solanaceae which is widely recognized for its beneficial properties and is used in Chinese herbal medicine. *Lycium barbarum* administered to STZ diabetic rats for eight weeks was observed to result in amelioration of retinal ERG [129], which was likely to be related to the strong anti-apoptotic activity of this herb as reported in a retinal ischemia/reperfusion model [130]. The anti-apoptotic action of lutein may be related to autophagy promoting effects of this carotenoid, as reported for both human retinal pigment epithelial ARPE-19 and mouse photoreceptor 661W cells exposed to cytotoxic stress [114].

### 4.4. Other Compounds

Treatment with Lisosan G restores expression of caspase 3 to control levels in ex vivo mouse retinal explants subjected to oxidative stress. In STZ diabetic rats, Lisosan G reduces neuronal death and favors an improvement in retinal functionality, as evaluated by ERG. This result indicates that treatment with Lisosan G is able to protect both the inner and outer retina from diabetes-induced alterations [63]. Other compounds with documented neuroprotective effects in models of DR include zerumbone, whose anti-apoptotic effects correlate with improvement of retinal histological alterations and reduction of retinal thickness in diabetic rats [90], and *Aralia elata*, which protects mouse retinas from diabetes-induced decreases in retinal thickness, increases in TUNEL labeled ganglion cells, and increases in active caspase-3 [94]. Finally, an anti-apoptotic function, although not a direct neuroprotective effect, has been be attributed to taurine, a non-essential free aminoacid found in *Lycium barbarum* which has been reported to inhibit high glucose-promoted caspase-3 expression and activity in ARPE-19 cells [131].

### 4.5. Relationships between Oxidative Stress, Inflammation, and Neurodegeneration

Most of the nutraceuticals cited above possess antioxidant, anti-inflammatory, and neuroprotective properties at the same time. It is unlikely that these capacities are expressed independently from each other. Rather, the evidence suggests that they are intimately correlated. Indeed, oxidative stress and ROS toxicity may lead directly to DNA and protein damage, but, as mentioned above, oxidative stress is also linked to inflammation. Both oxidative stress and inflammation, then, would be able to cause neurodegeneration. Treatments with antioxidant compounds in early phases of DR may represent an efficacious way to preserve the retina from further damage due to inflammation and from extensive neurodegeneration. In this sense, nutraceutical antioxidants may represent a novel class of compounds with interesting potential therapeutic value for DR [132].

## 5. Nutraceuticals and Vascular Changes

On the basis of vascular changes, DR is classified as a non-proliferative diabetic retinopathy (NPDR) or proliferative diabetic retinopathy (PDR). NPDR is characterized by microvascular damage including BRB breakdown, pericyte loss, acellular capillaries, capillary occlusion, and thickening of the basement membrane. In PDR, neoangiogenesis phenomena are observed and new blood vessels are generated. These vessels create a deleterious action in the retina because of their mechanic traction, which, in the end, causes retinal detachment and consequent blindness [133]. As outlined below, VEGF, acting at its main receptor vascular endothelial growth factor receptor-2 (VEGFR2), plays prominent roles in both phases of DR.

The BRB represents a filter allowing selective passage of substances from the bloodstream to the retina, thereby regulating osmotic equilibrium, ionic concentrations, and transport of nutrients. These functions are based on the presence of tight and adherens junctions between adjacent cells. Tight junctions are composed of proteins like occludin, claudin, and zonula occludens 1 (ZO-1). These proteins are the principal compounds implicated in BRB functionality, creating a strong bond between endothelial cells and regulating the transport of solutes and molecules through prevention of the unchecked diffusion of substances between the bloodstream and neuroretina [134]. In DR, oxidative stress and inflammation result in complex changes causing upregulation of cytokines and growth factors, among which VEGF is the most implicated in BRB dysfunctions [135,136]. Indeed, VEGF upregulation is correlated with alterations of the tight junction structure caused by VEGF-induced phosphorylation and downregulation of tight junction proteins (i.e., ZO-1 and occludin) [137,138]. In addition, overexpressed VEGF also induces phosphorylation of the adherens junction protein VE-cadherin, further favoring increased BRB permeability [139]. VEGF upregulation in DR also correlates with increased expression of intercellular cell adhesion molecule 1 (ICAM-1), which in turn promotes leucocyte adhesion and capillary occlusion [13]. Other cytokines and chemokines are implicated in BRB impairment. For instance, TNFα overexpression is associated with decreases in occludin, claudin, and ZO-1 expression, while IL-1β induces barrier dysfunction through leukocyte recruitment and release of the vasoactive amine histamine [140,141]. Matrix metalloproteinases (MMPs) play important roles both in the early stages of DR, when MMP-2 and MMP-9 promote the apoptosis of retinal capillary cells, and in the later phase, when they facilitate neovascularization by degrading the extracellular matrix [142].

Other early vascular pathological changes in NPDR include loss of pericytes and thickening of the basement membrane. Pericytes are contractile cells located at the surface of capillaries, implicated in blood vessel stability, blood flow regulation, and formation of the BRB. In NPDR, pericyte loss occurs even before endothelial injury and is directly correlated with accumulation of AGEs, impairment of the BRB, and vascular leakage [143,144]. Apoptosis of pericytes in NPDR also leads to formation of microaneurysms and acellular capillaries [145]. Thickening of the basement membrane, due to the increase in vascular basal membrane compounds such as laminin and collagen IV [136], may contribute to the disruption of the tight link between pericytes and endothelial cells, causing pericyte apoptosis, whereas the endothelium, deprived of proliferation control, can give rise to new vessels [146].

PDR is characterized by neovascularization coupled with fibrotic responses at the vitreoretinal interface, and subsequent blindness due to vitreous hemorrhage, retinal fibrosis, tractional retinal detachment, and neovascular glaucoma [147,148,149]. Out of all the angiogenesis regulators, VEGF has been most extensively studied and provides the basis for current anti-angiogenic therapy [150]. VEGF plays a crucial role in PDR pathogenesis by promoting neovascularization through binding to VEGFR2 expressed on endothelial cells, inducing endothelial cell proliferation and sprouting angiogenesis [151].

The protective actions of nutraceuticals against microvascular changes typical of NPDR have been investigated in a variety of DR models. However, these models do not reproduce the neoangiogenesis characterizing PDR, and evidence of possible antiangiogenic properties of nutraceuticals has been found in other experimental models favoring the growth of new retinal vessels, mainly rodents with oxygen induced retinopathy (OIR) or experimental choroidal neovascularization (CNV). Other indications of the possible antiangiogenic effects of nutraceuticals have been derived from observations of their efficacy in inhibiting endothelial cell proliferation, migration, and tube formation. A summary of the effects of nutraceuticals against vascular changes in models of DR or of neoangiogenesis is given in Figure 3.

### 5.1. Non-Flavonoid Polyphenols 

The vasoprotective potential of curcumin has been tested in vitro and in vivo. Treatment with curcumin prevents increases in glucose-induced VEGF expression as well as cellular proliferation in HRECs [30]. In addition, pre-treatment with curcumin has been shown to prevent capillary degeneration in rat retinas after ischemia reperfusion injury [152]. In diabetic rodents, curcumin has also been observed to protect pericytes from structural degeneration and to reduce VEGF expression, retinal vascular leakage, thickening of the basement membrane, vessel diameter, and vessel tortuosity [27,29,112,153]. Finally, curcumin has been reported to suppress experimental CNV and activation of hypoxia inducible factor 1α (HIF-1α, a transcription factor promoting VEGF expression and release) in mice [154].

In hypoxic ARPE-19 cells, resveratrol has been found to significantly inhibit HIF-1α and VEGF by blocking the PI3K/Akt/mTOR signaling pathway and by promoting proteasomal HIF-1α degradation [155]. Resveratrol also reduces diabetes-induced VEGF and ICAM-1 expression, leukocyte adhesion, pericytes loss, and prevents BRB breakdown as well as vascular leakage in the retinas of diabetic mice and rats [78,80,156,157]. In addition, extracts of *Polygonum cuspidatum*, containing resveratrol, have been shown to inhibit retinal vascular permeability and the loosening of the tight junctions in diabetic rats [79]. In mice with CNV induced by laser photocoagulation, resveratrol has been observed to significantly inhibit CNV growth [155,158] and reduce retinal neovascular lesions in very low-density lipoprotein receptor mutant mice, which are characterized by retinal neovascularization, by inhibiting VEGF expression as well as endothelial cell proliferation and migration [159]. The potential antiangiogenic effects of resveratrol and its possible use in DR treatments have been recently reviewed [160].

### 5.2. Flavonoid Polyphenols 

Many of the microvascular changes and angiogenesis processes that occur in DR are inhibited by treatment with different flavonoids. Quercetin has been reported to reduce VEGF and MMP-9 expression in the retinas of diabetic rats [161]. In experiments with the rhesus choroids-retina endothelial cell line RF/6A, quercetin has also been reported to inhibit VEGF-induced endothelial cell proliferation, migration, and tube formation, suggesting that it may efficiently inhibit choroidal or retinal neovascularization [162,163]. 

Chrysin has been found to ameliorate diabetes-mediated microvascular and neovascular abnormalities in studies with HRECs and with retinas of *db/db* mice. Indeed, it increases the stability between endothelial cells by increasing ZO-1 and VE-cadherin expression and reduced vascular permeability and vasoregression. Chrysin also restricts the phenomena of neovascularization and prevents the onset of neovascular tufts. Its actions are likely to be mediated by inhibition of the upregulated HIF-1α-VEGF-VEGFR2 axis [164]. In addition, intravitreally injected chrysin has been found to exert an inhibitory effect on CNV in an experimental rat model [165].

Among the green tea catechins, epigallocatechin gallate treatment of ARPE-19 cells reduces VEGF, VEGFR2, and MMP-9 mRNA expression and inhibits proliferation, vascular permeability, and tube formation in VEGF-induced human retinal microvascular endothelial cells (HRMECs). In addition, it also reduces BRB breakdown in VEGF-induced animal models [166]. Epicatechin has been reported to reduce apoptosis and AGE accumulation in retinal vascular cells of intravenously AGE injected rats [167]. Interestingly, green tea fractions have been reported to decrease neovascularization in the OIR rat model; however, the active components of green tea displaying such effects do not seem to contain catechins [168]. 

Hesperetin in diabetic rats has been found to inhibit VEGF expression, decrease vascular permeability and leakage, and restore the normal thickness of the basement membrane [169]. Another flavanone compound, naringenin, has been reported to attenuate laser-induced CNV in rats [170], an effect that is increased if naringenin is complexed with β-cyclodextrin, which improves naringenin water solubility [171]. Belonging to the same class of flavonoids as hesperetin and naringenin, eriodictyol has been described as being able to lower the retinal levels of VEGF, ICAM-1, and endothelial nitric oxide synthase, which is involved in BRB breakdown, in STZ rat retinas [85].

Both the flavone glycoside baicalin, found in several plant species of the genus *Scutellaria*, and the natural flavone luteolin, abundantly present in several plant products, including broccoli, pepper, thyme, and celery, display antiangiogenic properties in models of retinal neovascularization. Indeed, intravitreally-injected baicalin inhibits the growth of CNV in rats [172], while intravitreal luteolin has been reported to inhibit retinal neovascularization in the mouse OIR model and to suppress hypoxia-induced VEGF expression (via inhibition of HIF-1α) as well as VEGF-induced migration and tube formation in HRMECs [173]. Similarly to baicalin and luteolin, deguelin, a derivative of the isoflavonoid rotenone and a naturally occurring insecticide isolated from plants of the *Mundulea sericea* family, effectively reduces both CNV and OIR neovascularization [174,175]. It has also been shown to inhibit tube formation of human umbilical vein endothelial cells (HUVECs) and in vivo angiogenesis of chick chorioallantoic membrane [174], which is consistent with deguelin antiangiogenic activity. In addition, deguelin analogs have been recently produced which inhibit HIF-1α and reduce both in vitro angiogenesis and neovascularization in the OIR model [176].

In line with the other flavonoids cited above, the naturally occurring homoisoflavonoids cremastranone and homoisoflavanone, which are both found in *Cremastra appendiculata*, traditionally known as a medicinal plant in East Asia, have been observed to reduce both CNV and neovascularization in the OIR model, and to inhibit HMREC or HUVEC proliferation, migration, and tube formation [177,178].

Chalcones are natural compounds which are present in edible plants. Intraperitoneal administration of trans-chalcone in a mouse OIR model has been shown to significantly inhibit neovascularization and VEGF as well as ICAM-1 upregulation [179]. In addition, intravitreal administrations of isoliquiritigenin, from licorice root, have been observed to alleviate neoangiogenesis in both the CNV and the OIR models, and suppress neovascularization in the corneal neovascularization assay and VEGF-induced vessel growth in an *ex ovo* chick chorioallantoic membrane assay [180].

Blueberry anthocyanins are very effective in preventing the onset of microvascular damage. In the retinas of diabetic rats treated with *Vaccinium myrtillus* extracts, VEGF levels have been seen to be reduced, the expression of the tight junction proteins claudin-5, occludin, and ZO-1 is restored, and BRB breakdown is prevented [181].

### 5.3. Carotenoids

Dietary lutein has been shown recently to promote a decrease in the extent of CNV induced by laser photocoagulation in mice. This effect increases in an additive manner when lutein is administered together with ω-3 long-chain polyunsaturated fatty acids and it is accompanied by reductions in oxidative stress and in inflammatory mediators [182].

### 5.4. Saponins

Rk1 ginsenoside, a derivative of natural ginseng, has been implicated in the prevention of pathological loss of vascular integrity thanks to its strong anti-vascular permeability action. Rk1 ginsenoside reduces leakage of retinal vessels in diabetic mice and, in HRMECs, inhibits endothelial permeability caused by VEGF and other vasoactive factors such as thrombin and histamine [183]. Ginsenoside Re has also been reported to exert protective effects against vascular damage in the retinas of diabetic rats [184].

### 5.5. Other Compounds

In retinas of STZ rats, Lisosan G prevents VEGF upregulation and VEGFR2 stimulation, as demonstrated by reduction of VEGFR2 phosphorylation. Consequently, the diabetes-induced reduction of occludin and ZO-1 expression is also inhibited by Lisosan G. These effects result in protection of the BRB, as evidenced by a dramatic reduction in vascular leakage in the retinas of STZ rats treated with Lisosan G with respect to the retinas of control STZ rats [63]. 

Acetyl-11-keto-β-boswellic acid (AKBA) is an active principle derived from the plant *Boswellia serrata*. It has been found to efficiently inhibit pathologic neovascularization in a mouse OIR model. AKBA inhibits upregulation of VEGF expression, which is typical of OIR, likely by affecting the Src homology region 2 domain-containing phosphatase 1/signal transducer and activator of transcription 3/VEGF axis [185].

*Osteomeles schwerinae* C. K. Schneid (Rosaceae) is a native plant in Asia. An ethanolic extract of this plant, referred to as K24, has an inhibitory effect on AGE-induced retinal vascular leakage by suppressing the expression of VEGF and decreasing occludin downregulation. In addition, K24 inhibits neovascular growth in retinas of OIR mice [186].

Extracts of *Zingiber officinale* orally administered to diabetic rats result in the normalization of the retinal vessel diameter and reduction of basement membrane thickness [89]. Diabetes-induced BRB breakdown is prevented with extracts of *Zingiber zerumbet* rhizome, containing principally kaempferol, quercetin, curcumin, and zerumbone. An ethanol extract of the rhizome administered to diabetic rats reduces vascular permeability and vessel dilation, favors an increase in tight junction protein expression, reduces VEGF and pro inflammatory molecule expression, causes a decrease in adhesion molecules such as ICAM-1, and alleviates leukostasis [187]. The vasoprotective effect of ginsenosides is also observed when they are in combination with other compounds. For instance, *Panax notoginseng* may be combined with other Chinese herbs, such as *Salvia miltiorrhiza, Astragalus membranaceus,* and *Scrophularia ningpoensis*, to generate a compound called Fufang Xueshuantong, which causes an improvement in microvascular lesions, induces decreases in VEGF and ICAM-1 expression and BRB breakdown together with an increase in occludin expression [188,189]. Similarly, adding *Panax notoginsen* to Dang Gui Bu Xue Tang, an aqueous extract of *Radix Astragali* and *Radix Angelica sinensis* used in traditional Chinese medicine, reduces VEGF levels, occludin expression, vascular permeability, leukostasis, and the number of acellular capillaries in the retinas of diabetic rats [190].

Another extract that may reduce vascular damage in DR is the fortified extract of red berries, *Ginkgo biloba*, and white willow bark, as cited above. Indeed, in addition to inhibition of TNFα levels, it also induces attenuation of VEGF upregulation in the retinas of STZ rats [95].

### 5.6. Relationships between Oxidative Stress, Inflammation, Neurodegeneration, and Vascular Damage

As discussed above, nutraceuticals display neuroprotective effects due to their antioxidant and anti-inflammatory properties, as demonstrated in different experimental models of DR. These same compounds, or compounds that have been demonstrated to possess antioxidant and/or anti-inflammatory properties in other models, also protect the retina from the vascular damage and vascular proliferation typical of DR. It is interesting to note that in studies analyzing VEGF in DR models after treatment with neuroprotectants, decrease in apoptotic markers is always associated with a decrease in VEGF expression and/or release (see for instance [12,63]). These observations can be explained by assuming that those compounds also exert an independent regulation of the VEGF biosynthetic pathways or of the cell response to VEGF, as suggested by the observed effects of nutraceuticals on VEGF-induced endothelial cell proliferation, migration, and tube formation, or in models of retinal neoangiogenesis. However, the existence of a causal relationship between neuronal damage and vascular responses is a more likely hypothesis. Therefore, the effects of diabetes in the retina may include an initial high glucose-induced oxidative stress that elicits an inflammatory response and provokes damage of neurons and of other retinal cells. Neuronal suffering then would trigger expression and release of VEGF, mainly from Müller cells, which would act as a neuroprotective factor. Indeed, VEGF has recognized neuroprotective properties, but its prolonged upregulation will induce microvascular damage, BRB breakdown, and, in the long term, neoangiogenesis [3]. The assumption of nutraceuticals from the earliest evidence of diabetes will strengthen the antioxidant power in the retina, reducing oxidative stress and inflammation, with consequent protection from cell death, absence of VEGF upregulation, and no induction of vascular changes (Figure 4).

## 6. Clinical Studies

There are only a few clinical studies investigating the possible use of nutraceuticals for the treatment of DR, and most of them have been focused on carotenoids. Randomized clinical trials in patients with NPDR have shown that supplementation with lutein for three or for nine months results in increased visual acuity and contrast sensitivity, while foveal thickness decreases, indicating an alleviation of macular edema [191,192]. Similar results have been obtained in a placebo-controlled randomized clinical trial with patients affected by diabetic maculopathy refractory to conventional therapy, in which administration of 15 mg crocin tablets per day for three months caused a significant improvement of both best-corrected visual acuity and central macular thickness [193]. In addition, Type 2 diabetes patients having a higher ratio of serum non-pro-vitamin A carotenoids (lutein, zeaxanthin, lycopene) to pro-vitamin A carotenoids (α-carotene, β-carotene and β-cryptoxanthin) have shown a 66% reduction in risk for DR [194]. Moreover, the optical density of the macular pigment, which comprises the carotenoids lutein and zeaxanthin [49], has been reported to be lower in patients with Type 2 diabetes than in age-matched controls, and still lower in patients with Type 2 diabetes and DR [195]. Finally, a retrospective study with Type 2 diabetic patients after two years of carotenoid supplementation has suggested that carotenoids may have a beneficial effect on the macular function of diabetic patients [196].

In addition to clinical studies on carotenoids, there are also a few papers which have reported the use of other nutraceuticals in patients suffering from DR. For instance, a standardized phytosomal curcuminoid mixture (Meriva^®^) greatly improves curcumin absorption [197], and in one study 38 diabetic patients treated with Meriva_®_ showed improvements in diabetic microangiopathy and retinopathy at four weeks post-treatment [198]. In addition, a recent study has investigated potential beneficial effects of green tea. Indeed, a clinic-based, case-control study performed on diabetic patients with Type 2 diabetes showed that those who regularly drank Chinese green tea every week for at least one year in their lives had a DR risk reduction of about 50% compared with those who had not [199].

## 7. Bioavailability of Nutraceuticals

Bioavailability is a pharmacokinetic term referring to the fraction of bioactive compound that reaches the blood circulation without undergoing alterations. The index of bioavailability of nutraceuticals is important because it allows for the calculation of the right dose of nutraceutical to ingest. For this reason, understanding the oral bioavailability of a nutraceutical compound is as important as understanding its therapeutic potential. After ingestion, botanical compounds must overcome a series of threats that may alter their structure before they can reach systemic circulation, for instance, the environment of the gastrointestinal tract and the intestinal as well as the hepatic metabolism. Unfortunately, many nutraceuticals have low oral bioavailability, and therefore investigations to improve this aspect are of fundamental importance. Recently, significant steps forward have been made to develop new technologies using analogous compounds, nanoformulations, or nanoparticles, which may protect the nutraceutical from enteric adverse conditions [200,201,202,203]. 

Curcumin is characterized by poor bioavailability mainly due to low solubility, rapid metabolism and poor absorption, which, despite its medical efficacy, limits its clinical applications [204]. Conjugation of curcumin to metal oxide nanoparticles or encapsulation in lipid nanoparticles, dendrimers, nanogels, or polymeric nanoparticles, improves the water solubility and bioavailability of curcumin, thus increasing its pharmacological effectiveness [76]. The encapsulation of curcumin in the calix [4] arene nanoassembly limits curcumin degradation and increases its solubility, enhancing the effect of the compound on antioxidant and anti-inflammatory markers in both in vivo and in vitro models [205]. Similar results have been obtained using a different nanocarrier formulation comprising Pluronic-F127 stabilized d-α-Tocopherol polyethene glycol 1000 succinate nanoparticles [206]. A recent study has reported that, among different tested curcumin formulations, only that containing a hydrophilic carrier may provide therapeutic levels of curcumin in rabbit retinas [32].

Resveratrol, similarly to curcumin, is known for its poor oral bioavailability and scarce pharmacokinetic properties due to low aqueous solubility and low photostability, which compromise its great potential. In fact, as shown by pharmacokinetic studies, the levels of unmetabolized resveratrol after oral administration are reduced to about 1% due to its high intestinal and hepatic metabolism [207]. To solve this problem, different resveratrol nanoformulations have been tested, including liposomes, solid lipid nanoparticles, polymeric nanoparticles, and cyclodextrins. The use of these alternative administration methods generates different advantages because they improve solubility, bioavailability, and physical chemical stability, and favor a controlled drug release [208,209,210]. The use of resveratrol analogs could be another alternative choice for administration of this nutraceutical. The pharmacokinetic profiles of resveratrol and its analog perolstilbene have been analyzed in rats, showing that the bioavailability of perolstilbene was 80% and that of resveratrol 20% [211]. A summary of oral delivery systems for resveratrol has recently been published [160].

Nanoparticles can also be used to increase the bioavailability of epigallocatechin gallate, another nutraceutical characterized by low solubility and stability. Different nanosystems have been used for epigallocatechin gallate delivery, including liposomes, gold nanoparticles, inorganic nanocarriers, and lipid as well as polymeric nanoparticles [212,213].

A recent study has reported that the distribution of an orally administered nutraceutical may vary substantially depending on tissue type. Indeed, in a pilot study, ^13^C-lutein was detected in a variety of tissues in a rhesus macaque after a single oral administration, but not in the retina [214]. Some improvement in lutein delivery to ocular tissues may derive from lutein encapsulation into hyaluronic acid-coated PLGA nanoparticles, which have been demonstrated to efficiently bind ARPE-19 cells and improve the physicochemical properties of lutein [215].

## 8. Conclusions

A review of the effects produced by the administration of nutraceuticals in DR-related models indicates that all of the pathologic conditions seen in DR, including oxidative stress, inflammation, neurodegeneration, and vascular lesions can be alleviated by many of these natural compounds. There is evidence suggesting that oxidative stress, induced by diabetes through different pathways, might promote inflammation and cause neurodegeneration. Neuronal suffering, in turn, would trigger VEGF upregulation, causing subsequent vascular damage. Therefore, it appears that an increased antioxidant defense, if established before extended neuronal and vascular lesions, could reduce the subsequent pathological changes. A continuous supplementation of nutraceuticals with diet could afford a sufficient antioxidant power, and nutraceutical-based approaches may be the most efficacious, economic, and sustainable treatments to limit or even prevent the development of DR in diabetic subjects.

Despite this attractive perspective, however, clinical studies examining the real potential of nutraceuticals to ameliorate DR are still very limited in number. This is probably due the fact that it is not totally clear whether nutraceuticals should be tested to treat DR or to prevent DR. The evidence reported in this review has led us to hypothesize a chain of events (see Figure 4) that could be prevented by nutraceuticals; nutraceuticals may not be as efficient in treating the disease once it has been established. When investigating the preventative value of nutraceuticals, clinical studies are probably more difficult to organize and would require considerably long time periods.

Another reason for limiting clinical studies is likely the poor bioavailability of most nutraceuticals. As long as efficient delivery methods are not available for nutraceuticals to exert significant biological action in the retina, it will be difficult to design meaningful clinical investigations. Studies investigating new strategies for nutraceutical delivery, mainly based on nanoformulations, are very recent (they have appeared in the last ten years), and hopefully in the near future new research may fill this gap and promote new clinical experimentation of nutraceuticals. 

## Figures and Tables

**Figure 1 nutrients-11-00771-f001:**
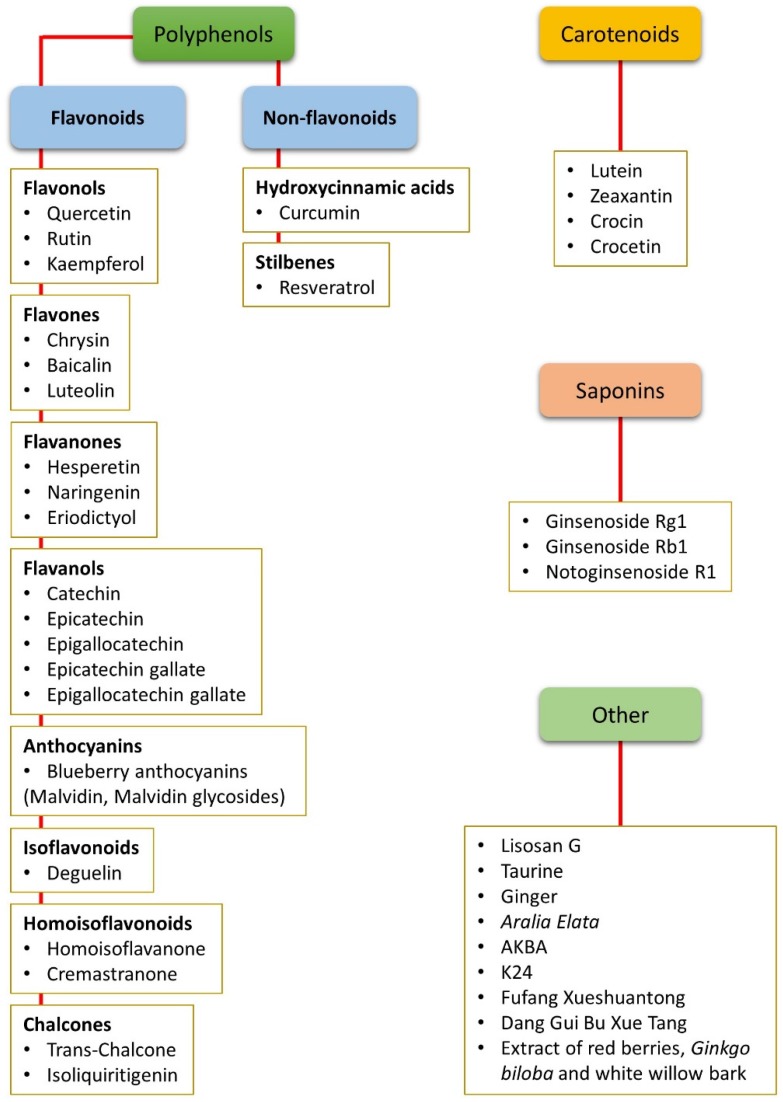
Summary of all the nutraceuticals cited in the present review. The compounds are listed according to their chemical classes, including polyphenols (both flavonoids and non-flavonoids), carotenoids, and saponins. Other compounds that do not belong to any of these classes or that are mixtures of different chemicals are classified as “other”. AKBA: Acetyl-11-keto-β-boswellic acid.

**Figure 2 nutrients-11-00771-f002:**
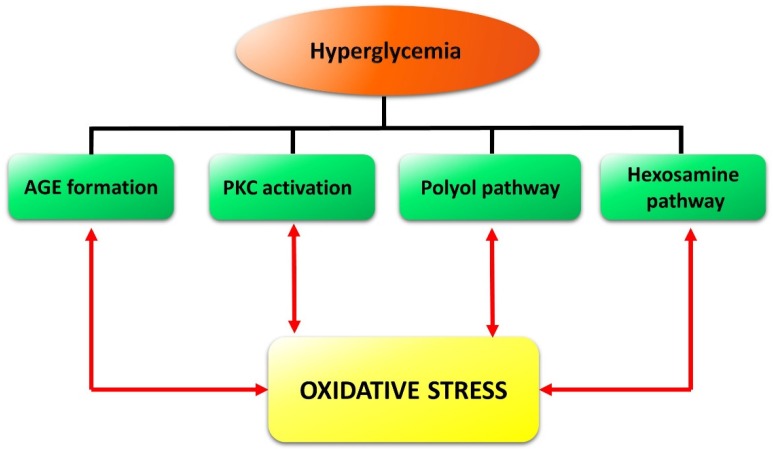
Schematic reconstruction of the events triggered in the retina by hyperglycemia and reinforced by oxidative stress in a vicious cycle. Formation of advanced glycation end-products (AGE) as well as the activation of protein kinase C (PKC), of the polyol pathway, and of the hexosamine pathway, are the main diabetes-induced abnormalities related to diabetic retinopathy.

**Figure 3 nutrients-11-00771-f003:**
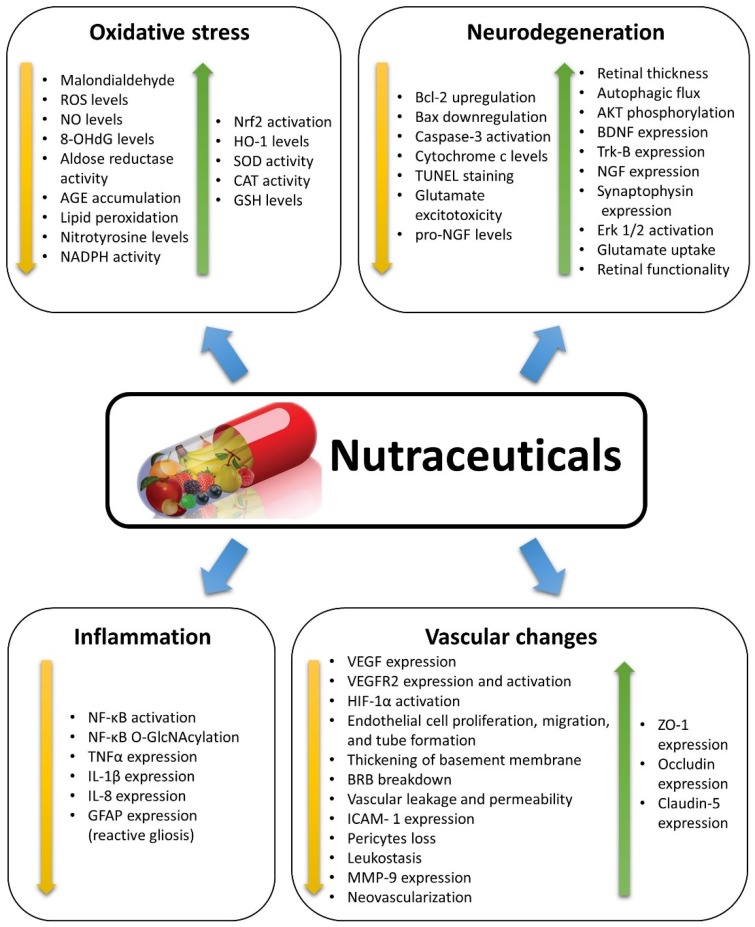
Summary of the effects induced by nutraceuticals as described in the studies reviewed herein. Nutraceuticals exert positive effects in diabetic retinopathy, counteracting the diabetes-induced changes by decreasing (yellow arrows) or increasing (green arrows) the expression/activation of specific factors or the occurrence of some events. 8-OHdG: 8-hydroxy-2′-deoxyguanosine; AGE: Advanced glycation end-products; AKT: Protein kinase B; Bax: Bcl-2-associated X protein; Bcl-2: B cell lymphoma 2; BDNF: Brain-derived neurotrophic factor; BRB: Blood-retina barrier; CAT: Catalase; Erk 1/2: Extracellular signal-regulated kinase 1/2; GFAP: Glial fibrillary acidic protein; GSH: Glutathione; HIF-1α: Hypoxia inducible factor 1α; HO-1: Heme oxygenase-1; ICAM-1: Intercellular cell adhesion molecule 1; IL-1β: Interleukin 1 beta; MMP-9: Matrix metalloproteinase-9; NADPH: Nicotinamide adenine dinucleotide phosphate; Nf-kB: Nuclear factor kappa-light-chain-enhancer of activated B cells; O-GlcNAc: O-linked β-N-acetylglucosamine; NGF: Nerve growth factor; NO: Nitric oxide; Nrf2: Transcription nuclear factor erythroid-2-related factor-2; ROS: Reactive oxygen species; SOD: Superoxide dismutase; TNFα: tumor necrosis factor alpha; Trk-B: Tyrosine receptor kinase B; TUNEL: Terminal deoxynucleotidyl transferase-mediated dUTP nick end labelling; VEGF: Vascular endothelial growth factor; VEGFR2: Vascular endothelial growth factor receptor 2; ZO-1: Zonula occludens 1.

**Figure 4 nutrients-11-00771-f004:**
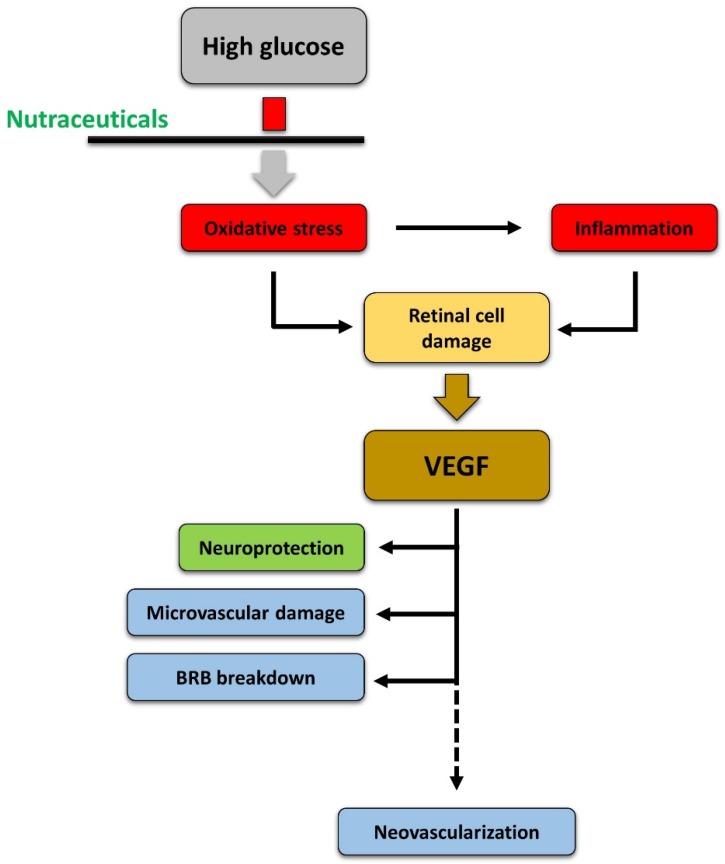
Hypothetic cascade of events induced by high glucose in the retina leading to diabetic retinopathy and the effects of nutraceuticals. See text for explanation.

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
