# Peer review of "Nutraceuticals for the Treatment of Diabetic Retinopathy"

_nutrients, 2019, doi:10.3390/nu11040771_

Round 1
Reviewer 1 Report
The first and most obvious issue is the precision of their English. I often, obviously, knew what they meant but the wording was subtly incorrect (scientific writing needs to be extremely accurate). For example, the second sentence of the abstract: …principal treatments for DR are represented by laser photocoagulation… That would be like saying that treatment for a broken arm is represented by a cast. It is not represented, per se, those simply are the treatment options. These kind of subtle English errors are throughout the manuscript (believe me, if I was writing something in Italian, the errors would be way worse).
It is also the case that the paper makes quite a few (mostly subtle, but not all minor) scientific errors. For example on Page 2, lines 67-72: Nutraceuticals can have other mechanistic properties other than simply antioxidants and anti-inflammatory. Those are not even the only way they act as antioxidants and anti-inflammatories. For example, carotenoids are not radical scavengers, they quench reactions and then dissipate the energy as heat. You also write “Nutraceuticals may be natural dietary supplements, and therefore they do not induce collateral side effects…” (recall that you start off the section defining nutraceuticals as food or part of a food). Of course, when food components are made into purified (often concentrated) dietary supplements, they can of course induce collateral side effects. There is a very long list that have been regulated for this very reason.
Finally, the overall organization of the manuscript is often a bit confusing. For example, the abstract seems to confuse prevention and treatment. It starts out with the somewhat dramatic treatments that are used to treat some of the more pernicious aspects of late-stage disease such as neovascularization. The segue then, however, is to look at food components as a means to prevent the disease. I think that most people agree that preventing degenerative diseases is the first and best means of reducing visual loss across the population. If someone, however, already has significant disease, the time for prevention has passed and more invasive treatment is really all that can be done (nutraceuticals at that point are not likely to make much difference). The abstract is basically expressing the choice as terrible invasive treatment or dietary intervention which is really a false dichotomy.
Since this is a review (not an empirical study), the writing, clarity, and saliency of the points is of paramount importance. This paper needs to be pretty extensively edited for grammar and accuracy.
Author Response
The first and most obvious issue is the precision of their English. I often, obviously, knew what they meant but the wording was subtly incorrect (scientific writing needs to be extremely accurate). For example, the second sentence of the abstract: …principal treatments for DR are represented by laser photocoagulation… That would be like saying that treatment for a broken arm is represented by a cast. It is not represented, per se, those simply are the treatment options. These kind of subtle English errors are throughout the manuscript (believe me, if I was writing something in Italian, the errors would be way worse).
R - We are aware that our English is far from perfect, and it cannot be otherwise since we are not native English speakers. I have to say that we have never received such a low score in English in our previous publications (and also the other Reviewers of this manuscript seemed to appreciate the language and the style). In any case, to correct possible “subtle” errors, we had the manuscript revised by an academic expert in English.
It is also the case that the paper makes quite a few (mostly subtle, but not all minor) scientific errors. For example on Page 2, lines 67-72: Nutraceuticals can have other mechanistic properties other than simply antioxidants and anti-inflammatory. Those are not even the only way they act as antioxidants and anti-inflammatories. For example, carotenoids are not radical scavengers, they quench reactions and then dissipate the energy as heat.
R- We agree that nutraceuticals can have multiple mechanistic properties, but the focus of this review paper is to consider nutraceuticals in the context of possible treatments of diabetic retinopathy, and the large majority (if not all) of the studies reviewed herein report antioxidant and / or anti-inflammatory actions of nutraceuticals. Therefore, we thought that a discussion of the multiple mechanisms of action associated to the effects of nutraceuticals was off topic.
We are aware that carotenoids can inactivate free radicals by quenching reactions. Different papers describe this property of carotenoids as a way in which they may act as radical scavengers (for instance PMID: 28546947 or PMID: 25134454) and we thought it was unnecessary to detail specific characteristics of selected classes of nutraceuticals. In any case, for the sake of precision, we have modified that paragraph of the Introduction.
You also write “Nutraceuticals may be natural dietary supplements, and therefore they do not induce collateral side effects…” (recall that you start off the section defining nutraceuticals as food or part of a food). Of course, when food components are made into purified (often concentrated) dietary supplements, they can of course induce collateral side effects. There is a very long list that have been regulated for this very reason.
R – This is a correct observation; however it could be applied to virtually all the compounds (of natural or synthetic origin) used for the treatment of any disease. The effect depends on the dosage and very high doses are usually detrimental. We thought it was unnecessary to specify it. We have added a brief note within brackets to clarify this aspect.
Finally, the overall organization of the manuscript is often a bit confusing. For example, the abstract seems to confuse prevention and treatment. It starts out with the somewhat dramatic treatments that are used to treat some of the more pernicious aspects of late-stage disease such as neovascularization. The segue then, however, is to look at food components as a means to prevent the disease. I think that most people agree that preventing degenerative diseases is the first and best means of reducing visual loss across the population. If someone, however, already has significant disease, the time for prevention has passed and more invasive treatment is really all that can be done (nutraceuticals at that point are not likely to make much difference). The abstract is basically expressing the choice as terrible invasive treatment or dietary intervention which is really a false dichotomy. Since this is a review (not an empirical study), the writing, clarity, and saliency of the points is of paramount importance. This paper needs to be pretty extensively edited for grammar and accuracy.
R – It was not our intention to propose a choice between invasive treatments and dietary intervention. We just wanted to express the idea that these invasive treatments may be prevented if appropriate administration of nutraceuticals begins in the early phases of the disease. This concept is probably better expressed in the introduction. We have rephrased the abstract to clarify this point.
Reviewer 2 Report
In this review paper, the authors summarise the role of nutraceuticals for the treatment of diabetic retinopathy. The review is clear and well-written. The paper will be of interest to readers of Nutrients. I have some comments that the authors may wish to address in order to strengthen the manuscript.
The end of Introduction section could benefit with a stronger last paragraph, which is currently rather vague (e.g. line 77: “Some results obtained in clinical studies are also presented”). The review is structured to be split into a number of sections, each with a different focus e.g. 3. Nutraceuticals and inflammation; 4. Nutraceuticals and neurodegeneration; 5. Nutraceuticals and vascular changes. The end of the Introduction might be strengthened by the addition of a brief description of these sections, helping the reader by providing an overview of how the review will be structured and the content of the rest of the paper.
The Conclusions section feels rushed and is not particularly helpful in its current format. It could be significantly improved by summarising the main findings and conclusions to be drawn from the evidence presented throughout the review. For example, the findings or the overall message of the Clinical Studies (Section 6) are not mentioned at all. Given that the main text states that there are only a few clinical studies investigating the possible use of nutraceuticals for treatment of DR, perhaps the lack of clinical studies in this area could be highlighted in the Conclusions section. Suggestions for future research, and highlighting gaps in current knowledge, would also strengthen this section.
Author Response
In this review paper, the authors summarise the role of nutraceuticals for the treatment of diabetic retinopathy. The review is clear and well-written. The paper will be of interest to readers of Nutrients. I have some comments that the authors may wish to address in order to strengthen the manuscript.
The end of Introduction section could benefit with a stronger last paragraph, which is currently rather vague (e.g. line 77: “Some results obtained in clinical studies are also presented”). The review is structured to be split into a number of sections, each with a different focus e.g. 3. Nutraceuticals and inflammation; 4. Nutraceuticals and neurodegeneration; 5. Nutraceuticals and vascular changes. The end of the Introduction might be strengthened by the addition of a brief description of these sections, helping the reader by providing an overview of how the review will be structured and the content of the rest of the paper.
R – This is a good suggestion. We have changed the last paragraph of the Introduction according to the Reviewer’s indications.
The Conclusions section feels rushed and is not particularly helpful in its current format. It could be significantly improved by summarising the main findings and conclusions to be drawn from the evidence presented throughout the review. For example, the findings or the overall message of the Clinical Studies (Section 6) are not mentioned at all. Given that the main text states that there are only a few clinical studies investigating the possible use of nutraceuticals for treatment of DR, perhaps the lack of clinical studies in this area could be highlighted in the Conclusions section. Suggestions for future research, and highlighting gaps in current knowledge, would also strengthen this section.
R – We appreciated this suggestion. Two paragraphs have been added to the Conclusions section to comment about the scarcity of clinical studies, its possible causes, and a possible perspective for future research.
Reviewer 3 Report
This was a very well written paper. I feel that this is a strong contribution to the field as the authors thoroughly review existing literature on nutraceuticals and DR. I though the summary paragraphs at the end of each section were exceptionally useful as well as the diagrams.
One minor comment - please clarify what is meant by "collateral side effects" on line 71.
I have no additional comments. This will be a nice contribution to the issue of nutrients and eye health.
Author Response
This was a very well written paper. I feel that this is a strong contribution to the field as the authors thoroughly review existing literature on nutraceuticals and DR. I though the summary paragraphs at the end of each section were exceptionally useful as well as the diagrams.
One minor comment - please clarify what is meant by "collateral side effects" on line 71.
I have no additional comments. This will be a nice contribution to the issue of nutrients and eye health.
R – Thank you. A couple of examples of possible collateral side effects have been added.